# ViP: A Differentially Private Foundation Model for Computer Vision

## Abstract

Artificial intelligence (AI) has seen a tremendous surge in capabilities thanks to the use of *foundation models* trained on internet-scale data. On the flip side, the uncurated nature of internet-scale data also poses significant privacy and legal risks, as they often contain personal information or copyrighted material that should not be trained on without permission. In this work, we propose as a mitigation measure a recipe to train foundation vision models via self-supervised learning with differential privacy (DP) guarantee. We identify masked autoencoders as a suitable learning algorithm that aligns well with DP-SGD, and train *ViP*—a **Vi**sion transformer with differential **P**rivacy—under a strict privacy budget of $\epsilon = 8$ on the LAION400M dataset. We evaluate the quality of representation learned by ViP using standard downstream vision tasks; in particular, ViP achieves a (non-private) linear probing accuracy of $55.7\%$ on ImageNet, comparable to that of end-to-end trained AlexNet (trained and evaluated on ImageNet). Our result suggests that scaling to internet-scale data can be practical for private learning.

## 1 Introduction

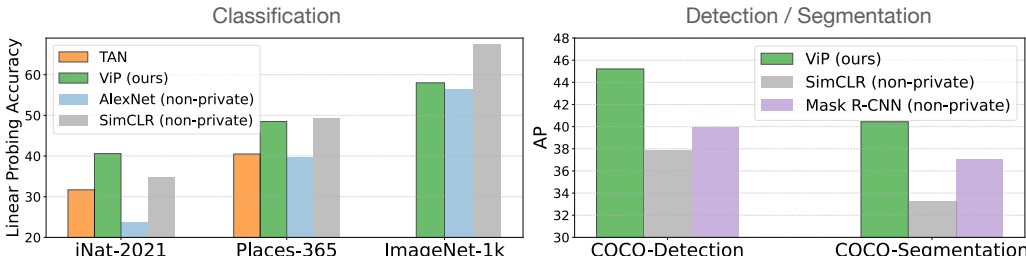

Figure 1: (**left**) Linear probing accuracies of TAN (Sander et al., 2022) (state-of-the-art DP training method), AlexNet (Krizhevsky et al., 2017), SimCLR (Chen et al., 2020a) and ViP—our DP-trained model with $\epsilon = 8$. ViP can achieve similar transfer learning result as SimCLR on iNat-2021 and Places-365, and achieves similar accuracy on ImageNet as end-to-end trained AlexNet. (**right**) Average precision (AP) evaluations of SimCLR (Chen et al., 2020a), Mask R-CNN (He et al., 2017) and ViP on MS-COCO. Our DP-trained model outperforms both SimCLR and Mask R-CNN.

Foundation models (*e.g.*, GPT-3, SimCLR, CLIP, *etc.* (Brown et al., 2020; Chen et al., 2020a; Radford et al., 2021)) pre-trained on vast amounts of diverse unlabeled data through self-supervised learning (SSL) have emerged as an important building block for artificial intelligence (AI) systems (Bommasani et al., 2021). These foundation models enable downstream applications via fine-tuning, prompting, or training a simpler model on top of the learned representations to perform more specialized tasks, and have performed tremendously well on challenging benchmarks in both language and vision domains (Brown et al., 2020; Radford et al., 2021; Touvron et al., 2023).

Despite the widespread deployment of foundation models, there are significant privacy and legal risks of training these models on uncurated data that often contain personal information or copyrighted material. Although the training data for these models are considered *public* in most cases, some of the data may be sensitive; additionally, there are certain privacy and copyright laws that apply to model training even on such *public* data (Henderson et al., 2023). In addition, studies have shown that generative foundation models such as GPT-3 can sometimes regurgitate memorized information about individuals and licensed content from its training data when prompted to do so (Carlini et al.,

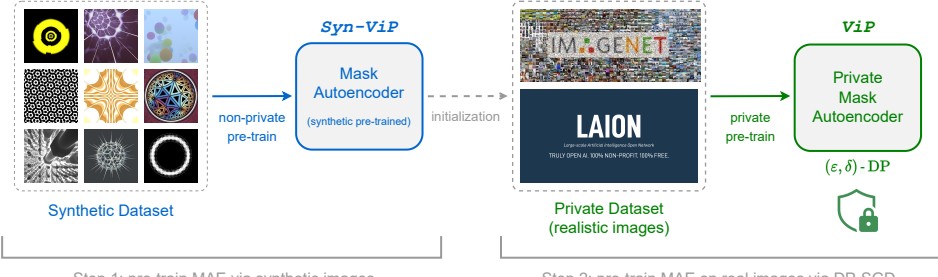

Figure 2: **How to pre-train differentially private transformers (*ViP*) with synthetic data?** In Step 1, we first pre-train a MAE model on synthetic images with standard optimizers (*e.g.*, SGD, AdamW). We denote this model by *(Syn)-ViP*. In Step 2, we use the MAE model pre-trained on synthetic images as initialization, and then apply differential private optimizers (*e.g.*, DP-SGD, DP-AdamW) to train a *ViP* model that satisfies $(\epsilon, \delta)$-DP.

2021). More recently, Meehan et al. (2023) showed that non-generative vision SSL models can also be probed to reveal sensitive information about individual samples in its training data when given partial information.

Given these risks, there is an urgent need to train foundation models that can adhere to relevant privacy and copyright laws. To this end, differential privacy (DP; Dwork et al. (2006)) seeks to limit the influence of individual training data points on the trained model, and hence has the potential to mitigate both privacy and copyright risks for sensitive information that is confined to a single or a few training examples (Henderson et al., 2023). For any model that can be trained using gradient-based optimization, DP-SGD (Song et al., 2013; Abadi et al., 2016) can be applied instead to ensure that the trained model satisfies the rigorous definition of DP. However, there are still significant technical challenges in DP-SGD training of large-scale foundation vision models:

1. Differentially private representation learning in general is a difficult problem. Tramer & Boneh (2020) showed that even handcrafted features can outperform feature learned by state-of-the-art DP-trained models, and attaining high-utility learned representations requires significantly more training data—much more than what is provided in typical supervised/curated datasets.

2. Combining self-supervised learning (SSL) with internet-scale *uncurated* datasets may seem like a natural approach to gain access to the large amount of data needed for DP training. However, most vision SSL training algorithms are based on *contrastive learning*, where the objective function depends on multiple samples in an entangled manner. This makes it difficult to perform the per-sample gradient computation needed in DP-SGD.

3. SSL training requires a much larger number of training epochs compared to supervised learning, which sharply increases the DP parameter $\epsilon$, leading to meaningless privacy guarantees.

In this paper, we describe a successful recipe for training differentially private large-scale foundation models via SSL. Firstly, we identify masked autoencoder (MAE; He et al. (2022)) as a promising SSL training algorithm that is amenable to DP-SGD. MAE uses an instance-separable loss function and does not require batch normalization, and hence per-sample gradients can be easily computed. We also show that it is tolerant to the large amount of Gaussian noise added in DP-SGD. Next, we demonstrate that MAE can effectively leverage synthetic datasets containing only programmatically-generated synthesized textures (Baradad et al., 2022) to warm-start the DP training process, significantly reducing the number of training epochs required to reach a high-utility model. The combination of these two ingredients forms a powerful DP training recipe for obtaining high-utility differentially private foundation vision models.

We implement this training recipe on the LAION400M dataset (Schuhmann et al., 2021). We show that the resulting model, which we call *ViP* (**Vi**sion transformer with differential **P**rivacy), learns highly useful and transferable representations—*rivaling that of representation learned by SimCLR on ImageNet*—while providing a strong DP guarantee with $\epsilon = 8$. In Figure 1, we compare ViP with other private and non-private models in terms of downstream linear probing accuracy and fine-tuning accuracy for different image datasets:

- For iNat-2021 and Places-365 classification, ViP outperforms both TAN (Sander et al., 2022)—the previous SOTA for DP supervised training—and AlexNet (Krizhevsky et al., 2017), while matching or exceeding the performance of SimCLR pre-trained on ImageNet.

- On ImageNet, the linear probing accuracy of ViP matches that of end-to-end trained AlexNet[1].

- On MS-COCO detection and segmentation, ViP outperforms both SimCLR pre-trained on ImageNet and Mask R-CNN.

Our experiments demonstrate that by scaling DP-SGD training to vast amounts of unlabeled data and using synthetic data to warm-start the model, we can attain high-utility foundation vision models under stringent privacy guarantees. Consequently, we hope that future work can continue to build on our successful recipe and further push the performance boundary of large-scale DP training.

## 2 BACKGROUND

**Differential privacy** (Dwork et al., 2014) is a mathematical framework for formal reasoning about information leakage through a private mechanism. A learning algorithm $\mathcal{A}$ is said to be $(\epsilon, \delta)$-*differentially private* (denoted $(\epsilon, \delta)$-DP) if for all training datasets $\mathcal{D}, \mathcal{D}'$ that differ[2] in a single training sample, we have:

$$P(\mathcal{A}(\mathcal{D}) \in S) \leq e^\epsilon P(\mathcal{A}(\mathcal{D}') \in S) + \delta \tag{1}$$

for all outcome sets $S$. More generally, equation 1 can be expressed as a statistical divergence $D(\mathcal{A}(\mathcal{D})||\mathcal{A}(\mathcal{D}'))$ between the distribution of models trained on $\mathcal{D}$ vs. $\mathcal{D}'$, with $(\epsilon, \delta)$-DP corresponding to the "hockey-stick" divergence (Sharma & Warsi, 2013). Another useful variant is *Rényi differential privacy* (RDP; (Mironov, 2017)), which uses the Rényi divergence $D_\alpha$ (Rényi et al., 1961): $\mathcal{A}$ is said to be $(\alpha, \epsilon)$-RDP if $D_\alpha(\mathcal{A}(\mathcal{D})||\mathcal{A}(\mathcal{D}')) \leq \epsilon$. Moreover, RDP can be converted to DP via the following (Balle et al., 2020): if $\mathcal{A}$ is $(\alpha, \epsilon_\alpha)$-RDP then it is also $(\epsilon, \delta)$-DP with

$$\epsilon = \epsilon_\alpha + \log\left(\frac{\alpha - 1}{\alpha}\right) - \frac{\log \delta + \log \alpha}{\alpha - 1}. \tag{2}$$

**DP-SGD training.** Abadi et al. (2016) showed that stochastic gradient descent (SGD)—the quintessential learning algorithm—can be made differentially private by perturbing the per-iteration gradient with Gaussian noise. The modified SGD update with gradient perturbation (often referred to as *DP-SGD*) is given by:

$$\boldsymbol{\theta}_{t+1} = \boldsymbol{\theta}_t - \frac{\eta_t}{|\mathcal{B}_t|} \left( \sum_{\mathbf{x} \in \mathcal{B}_t} \mathsf{clip}_C(\nabla_{\boldsymbol{\theta}} \ell(\mathbf{x}; \boldsymbol{\theta})|_{\boldsymbol{\theta}=\boldsymbol{\theta}_t}) + \mathcal{N}(0, \sigma^2 C^2 \boldsymbol{I}) \right), \tag{3}$$

where $\eta_t$ is the learning rate, $\mathcal{B}_t$ is the sampled batch, $\sigma > 0$ is the noise multiplier, and $\mathsf{clip}_C$ is the operation that clips the per-sample gradient norm to at most $C > 0$. It can be shown that this update procedure is $(\alpha, \epsilon_\alpha)$-RDP for some computable $\epsilon_\alpha$ (Mironov et al., 2019). The end-to-end learning algorithm by running $T$ iterations of SGD is thus $(\alpha, T\epsilon_\alpha)$-RDP via composition (Mironov, 2017), and a conversion to $(\epsilon, \delta)$-DP can be obtained using equation 2. Such privatization mechanism—per-sample clipping and injecting noise—can be easily integrated with other first-order optimization algorithms such as Adam (Kingma & Ba, 2014) and AdamW (Loshchilov & Hutter, 2017).

**Self-supervised learning (SSL)** has emerged as a prominent approach for scaling up the training of machine learning models to large-scale unlabeled datasets. Restricting our attention to the vision domain, SSL pre-trained models generalize effectively across a wide range of transfer learning downstream tasks such as classification, instance segmentation and object detection (Chen et al., 2020b; Bommasani et al., 2021), especially under the scenario of limited downstream training data. Vision SSL methods can be broadly categorized as either *joint embedding-based learning* (JE) (Chen et al., 2020a; He et al., 2020; Grill et al., 2020; Zbontar et al., 2021; Chen & He, 2021) or *reconstruction-based learning* (REC) (Bao et al., 2021; Xie et al., 2022; He et al., 2022). JE-based approaches design objective functions so that all views (or image augmentations) of the same sample have similar embeddings, while views of different samples have different embeddings. As a result, most JE-based approaches *require* a batch containing multiple samples in order to define the objective function. On the other hand, REC-based approaches aim to optimize models to reconstruct image inputs in the pixel space based on partially masked inputs, which promotes the model to learn compressed representations that can generalize well.

---

[1]The model is sourced from the PyTorch website and is end-to-end trained with supervised learning.

[2]We adopt the removal notion of adjacency, *i.e.*, $\mathcal{D}' = \mathcal{D} \cup \mathbf{z}$ for some $\mathbf{z}$ and vice versa.

**Related work.** Recently, an expanding body of literature has emerged on scaling DP training to large-scale datasets and models in both NLP and vision domains. In NLP, a series of works (Anil et al., 2021; Yu et al., 2021; Li et al., 2022a) showed that by combining public pre-training and scaling up the training batch size, it is possible to fine-tune the pre-trained language model to achieve reasonable downstream performance. In computer vision, Kurakin et al. (2022) first attempted to scale DP training of convolutional neural networks (ResNets) to ImageNet. De et al. (2022) further improved the performance of Kurakin et al. (2022) with a Normalizer-Free ResNet architecture and an improved training recipe. More recently, Sander et al. (2022) proposed a more efficient hyperparameter tuning method for DP training that led to state-of-the-art performance on ImageNet. It is worth noting that all these works on DP-trained computer vision models focus on training supervised models.

## 3 RECIPE FOR TRAINING DP FOUNDATION VISION MODELS

In this work, we identify a successful recipe for training differentially private foundation vision models. Training DP foundation models, or in general any deep learning model with a large number of parameters, poses a significant challenge due to the large amount of injected noise—$\mathcal{N}(0, \sigma^2 C^2 \boldsymbol{I})$ in equation 3. Indeed, current state-of-the-art differentially private deep learning models even underperform linear models with handcrafted features when $\epsilon$ is small (De et al., 2022; Tramer & Boneh, 2020). We propose two effective techniques that reduce the magnitude of noise injected during training while attaining strong $(\epsilon, \delta)$-DP guarantees: **1.** Scaling up the number of training samples via self-supervised learning with masked autoencoder; and **2.** Facilitating faster training by warm-starting the model with weights pre-trained on synthetic samples.

### 3.1 DIFFERENTIAL PRIVATE SSL WITH MASK AUTOENCODER

Most existing works on differentially private training (De et al., 2022; Sander et al., 2022; Bu et al., 2022) focus on supervised learning, which inherently restricts the quantity of training samples that can be utilized. In contrast, self-supervised learning approaches unlock the use of (albeit uncurated) internet-scale training data that can be on the order of billions of samples, which can potentially satisfy the amount of data needed for DP training of high-utility models (Tramer & Boneh, 2020).

On the other hand, most existing SSL training approaches do not align with requirements in DP-SGD training. For example, SimCLR (Chen et al., 2020a) requires a mini-batch of samples in order to compute the contrastive loss; BYOL (Grill et al., 2020) computes per-sample loss but it utilizes batch normalization (BN) (Ioffe & Szegedy, 2015) in the model architecture, resulting in each loss depending on a mini-batch of training samples.[3] Therefore, it is challenging to perform the per-sample gradient clipping as described in equation 3. Among various types of SSL methods, we identify reconstruction-base learning with masked autoencoders (MAE) (He et al., 2022) as one of the most suitable SSL approaches for training DP foundation vision models. The training objective $L_{\text{MAE}}(\boldsymbol{\theta})$ in MAE is defined as:

$$L_{\text{MAE}}(\boldsymbol{\theta}) := \frac{1}{n} \sum_{i=1}^{n} \ell_{\text{MSE}}(g \circ \psi(\text{mask}(\mathbf{x}_i); \boldsymbol{\theta}), \mathbf{x}_i) = \frac{1}{n} \sum_{i=1}^{n} \ell(\mathbf{x}_i; \boldsymbol{\theta}), \tag{4}$$

where $n$ is the number of training samples, $\mathbf{x}_i \in \mathbb{R}^{C \times H \times W}$ is the input of the $i$-th training image ($C$-number of channels, $H$-height, $W$-width), $\text{mask}(\cdot)$ is a function that mask out a fraction of the image, $\psi : \mathbb{R}^{C \times H \times W} \to \mathbb{R}^d$ is the encoder and $g : \mathbb{R}^d \to \mathbb{R}^{C \times H \times W}$ is the decoder. We use $\boldsymbol{\theta}$ to denote the trainable parameters of the $\psi$ and $g$, and use $\ell_{\text{MSE}}$ to denote the mean squared error (MSE) loss defined on the pixel space, *i.e.*, $\ell_{\text{MSE}}(\mathbf{x}_1, \mathbf{x}_2) = \|\mathbf{x}_1 - \mathbf{x}_2\|_F^2$. Similar to He et al. (2022), we apply vision transformers (Dosovitskiy et al., 2020) to instantiate the encoder and decoder maps. As shown in equation 4, the training objective can be decomposed into $n$ individual losses, and each individual loss $\ell(\mathbf{x}_i; \boldsymbol{\theta})$ only depends on the $i$-th training sample $\mathbf{x}_i$ and does not require the label of $\mathbf{x}_i$. Therefore, we can compute per-sample gradient $\nabla_{\boldsymbol{\theta}} \ell(\mathbf{x}_i; \boldsymbol{\theta})$ and perform per-sample gradient clipping without modifying the MAE training.

---

[3]Subsequent work by Richemond et al. (2020) demonstrated that BN can be substituted with group normalization by carefully modifying the model architecture. However, we have observed that the design of exponential moving averaged online network in BYOL can result in dynamic instability during training, which poses challenges in the context of DP training.

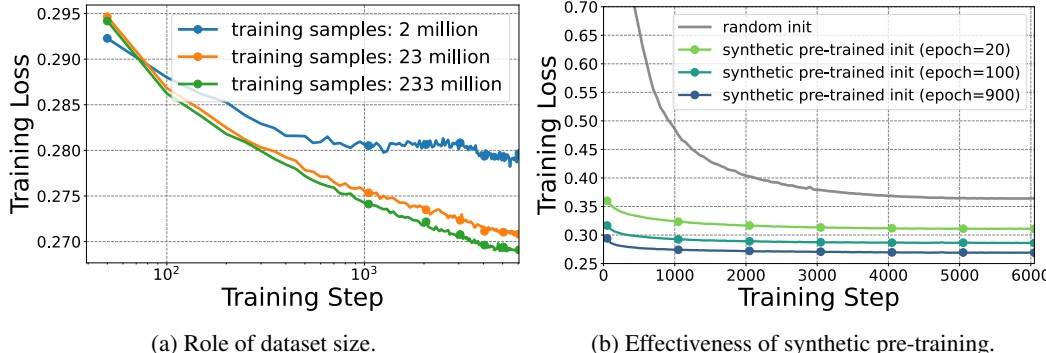

(a) Role of dataset size.    (b) Effectiveness of synthetic pre-training.

Figure 3: (a). We vary the number of training samples $n$ with the $(\epsilon, \delta_n)$-DP guarantee ($\delta_n = 1/2n$), and compare the training losses of MAE-DP. By scaling up the training dataset size, we can consistently improve the ViP training under the same $\epsilon$-DP budget. (b). Compared to ViP training from random initialization, we can significantly speed up the ViP training by leveraging the synthetic pre-trained MAE model as initialization.

By leveraging the self-supervised MAE training paradigm, we can now significantly scale up the training data size for DP SSL pre-training. Dataset scaling can effectively reduce the magnitude of noise in DP-SGD while maintaining the same $(\epsilon, \delta_n)$-DP guarantee, where $\delta_n = 1/2n$. As shown in Figure 3a, we investigate the impact of injected noise in ViP training by keeping all training hyperparameters the same except for the number of training samples[4]. With more training samples, the magnitude of the injected noise $\sigma$ becomes smaller. We find that when the noise magnitude is large, the training loss cannot be further optimized after certain number of training steps. In contrast, smaller magnitude of noise (as a result of larger training dataset) facilitates faster optimization of the training loss in comparison to larger noise scenarios. Importantly, the optimization trajectory is stable despite the presence of noise, allowing the MAE model to learn useful features.

### 3.2 SYNTHETIC PRE-TRAINING ENABLES FASTER DP TRAINING FOR VIP

Non-private training of SSL models often require a significant number of training epochs, much larger than what is required in supervised learning (Chen et al., 2020a; He et al., 2022; Balestriero et al., 2023). This creates an additional challenge for DP training since the number of training iterations $T$ directly impacts the privacy guarantee. Indeed, as mentioned in Section 2, DP-SGD with $T$ iterations is $(\alpha, T\epsilon_\alpha)$-RDP. Consequently, naively applying DP-SGD to MAE training results in an unfavorable privacy-utility trade-off.

Fortunately, He et al. (2019) demonstrated that using pre-trained initialization enables much faster model convergence compared to random initialization. However, in light of our discussion in Section 1, it is critical that the pre-training data does not contain any private information, even if the data is deemed "public". One promising alternative is pre-training on programmatically-generated synthetic images (Kataoka et al., 2020; Baradad et al., 2022), which was shown to achieve competitive downstream performance compared to pre-training on natural images. Doing so allows the MAE to learn spatial structure in the transformer modules (Jelassi et al., 2022) without expending any privacy budget for the natural image data. More importantly, synthetic pre-training does not carry any privacy risk, and legal risk is limited to obtaining proper license for the synthetic image generation code.

Thus, to accelerate ViP training, we pre-train the model on synthetic images generated using the Shaders21k tool developed in Baradad et al. (2022). Figure 2 shows samples of synthetic images generated by the tool. In Figure 3b, we compare the ViP training with and without synthetic pre-trained initialization. Notably, training ViP with synthetic pre-trained weights converges significantly faster than those with random initialized weights. Increasing the synthetic pre-training from 20 to 900 epochs further improves convergence for ViP training. Interestingly, as shown in Figure 1, MAE trained on the synthetic dataset already outperforms existing state-of-the-art DP-trained models (De et al., 2022; Sander et al., 2022) under our transfer learning evaluation, which shows that DP training on datasets even as large as ImageNet does not learn sufficiently expressive features (see Table 1).

---

[4]We maintain the same batch size across various data size settings while modifying the noise multiplier $\sigma$. Consequently, as the data size increases, the corresponding $\sigma$ values decrease.

### 3.3 OUR PROPOSED APPROACH

We now summarize our approach for DP foundation vision model training (also see Figure 2):

> **DP-MAES** – DP Masked Autoencoder with Synthetic Pre-training
>
> - **Step 1:** *Synthetic pre-training for initialization.* Pre-train mask autoencoder on the synthetic dataset with non-private optimizers.
> - **Step 2:** *DP training with synthetic initialization.* Apply the synthetic pre-trained model as initialization and train mask autoencoder on a large-scale natural image dataset (*e.g.*, LAION400M) with DP-SGD. The DP guarantee then applies to the natural image dataset.

It is worth mentioning that our proposed approach offers flexibility in the selection of both SSL training methods and synthetic datasets. For example, developing better synthetic datasets or more effective SSL learning method can further push the performance of the final DP foundation model.

## 4 EVALUATION

We evaluate the effectiveness of our training recipe by applying it to the LAION400M dataset to train our private foundation vision model: **ViP**. We consider various downstream tasks in order to demonstrate the quality and transferability of its learned representation. Furthermore, we compare ViP to previous state-of-the-art DP-trained models as well as widely adopted non-privately trained models, and find that ViP significantly improves SOTA for DP training on downstream transfer tasks (Section 4.2) and even outperforms non-private models on several challenging datasets. In addition to assessing the performance of ViP on non-private downstream tasks, in Section B.3, we also evaluate the ViP model via DP fine-tuning on ImageNet-1K, which shows a notable improvement of 10%+ absolute top-1 accuracy compared to previous SOTA (Sander et al., 2022). For additional experimental results on ViP, see Appendix B.

### 4.1 EVALUATION SETUP

Our implementation uses PyTorch, along with the functorch package (Horace He, 2021) for computation of per-sample gradients and the opacus package (Yousefpour et al., 2021) for privacy accounting. See Appendix A for additional implementation details.

**Datasets.** We use 1.05 million samples generated using the Shader21k (Baradad et al., 2022) tool as our synthetic pre-training dataset, and the LAION400M (Schuhmann et al., 2021) as our private pre-training dataset for the ViP model[5]. We evaluate ViP and baseline models via *non-private* linear probing and fine-tuning on the following downstream classification datasets: ImageNet-1K (Deng et al., 2009), Places-365 and Places-205 (Zhou et al., 2014), iNaturalist-2021 (Van Horn et al., 2021), CIFAR-100 (Krizhevsky et al., 2009), Caltech101 (Fei-Fei et al., 2006), and Aircraft (Maji et al., 2013). The input images are resized and center-cropped to 224×224 resolution. We also evaluate using MS-COCO instance segmentation and object detection (Lin et al., 2014), and semantic segmentation with the ADE20K dataset (Zhou et al., 2019) (in Appendix B.1).

**Model architecture.** Following He et al. (2022), we use vision transformer (ViT) (Dosovitskiy et al., 2020) to instantiate the masked autoencoder models. The default MAE-encoder has 12 transformer blocks and width 768, and the default MAE-decoder has 4 transformer blocks and width 512. We denote this MAE model as MAE-base. We also consider MAE models with different model sizes, including MAE-Nano, MAE-Tiny, MAE-Small and MAE-Large in Section 4.3.

**Optimization and hyperparameters for (DP-)MAE training.** We use AdamW (Loshchilov & Hutter, 2017) for training MAE – both for synthetic pre-training and differentially private MAE pre-training. When evaluating pre-trained models in downstream tasks, we apply LARS (You et al., 2017) for linear probing and AdamW for fine-tuning. For MAE training, we set the masking ratio to 75%. In terms of DP training, we set $\epsilon = 8.0$ and $\delta = 1/2n$ by default for training $(\epsilon, \delta)$-DP model. We set the clipping parameter $C = 0.1$, sampling ratio $q = 81920/n$, and noise parameter $\sigma = 0.5$.

---

[5]Some of the links in LAION400M are now broken since its initial release, and the version we use contains ~233 million real images. We use LAION233M to denote this subsampled version of LAION400M.

Table 1: Linear probing evaluation on downstream classification. We compare *ViP* with both private pre-training (DP-NFNet and TAN) and non-private pre-training (AlexNet and SimCLR) baselines, as well as the synthetically pre-trained MAE model: *(Syn)-ViP*. *ViP* consistently outperforms all private baselines, and has similar transfer learning performance as non-private SimCLR pre-trained on ImageNet-1K. ([‡]All models except for *(Syn)-ViP* and *ViP* are pre-trained on ImageNet-1K, giving them an unfair advantage for the linear probing evaluation on ImageNet-1K.)

| Model | DP? | SSL? | Pre-train dataset | # pre-train samples | ImageNet-1K[‡] | Places-365 | Places-205 | iNat-2021 |
|---|---|---|---|---|---|---|---|---|
| DP-NFNet | ✓ | ✗ | ImageNet-1k | ∼1 million | 45.3% | 40.1% | 39.2% | 28.2% |
| TAN | ✓ | ✗ | ImageNet-1k | ∼1 million | 49.0% | 40.5% | 38.2% | 31.7% |
| AlexNet | ✗ | ✗ | ImageNet-1k | ∼1 million | 56.5% | 39.8% | 35.1% | 23.7% |
| SimCLR | ✗ | ✓ | ImageNet-1k | ∼1 million | 67.5% | 46.8% | 49.3% | 34.8% |
| *(Syn)-ViP* | ✓ | ✓ | Shaders21k | ∼1 million | 49.8% | 43.2% | 45.8% | 32.4% |
| *ViP-LAION* | ✓ | ✓ | LAION | ∼233 million | **55.7%** | **46.1%** | **48.5%** | **38.1%** |
| *ViP-ImageNet* | ✓ | ✓ | ImageNet-1k | ∼1 million | 52.6% | 44.3% | 46.5% | 34.2% |

Table 2: Fine-tuning evaluation on few-shot downstream classification. ViP consistently outperforms both TAN (private) and AlexNet (non-private), as well as (Syn)-ViP by a large margin. Performance does fall short compared to non-private SimCLR pre-trained on ImageNet-1K despite having access to more than $100\times$ more data, suggesting that there is much room for improvement for private learning.

| Model | Aircraft | | | Caltech-101 | | | CIFAR-100 | | |
|---|---|---|---|---|---|---|---|---|---|
| | 10-shot | 20-shot | 30-shot | 5-shot | 10-shot | 30-shot | 5-shot | 10-shot | 30-shot |
| AlexNet | 23.27% | 34.47% | 41.35% | 64.70% | 73.57% | 81.40% | 29.74% | 36.31% | 49.28% |
| SimCLR | 38.79% | 56.90% | 64.90% | 81.70% | 89.11% | 94.51% | 49.93% | 60.18% | 71.84% |
| TAN | 22.84% | 37.93% | 46.01% | 49.32% | 66.42% | 77.87% | 21.28% | 27.78% | 42.35% |
| *(Syn)-ViP* | 21.79% | 46.85% | 58.45% | 60.51% | 76.21% | 88.48% | 27.62% | 38.96% | 55.84% |
| *ViP* | **31.62%** | **53.05%** | **64.26%** | **68.05%** | **79.03%** | **88.90%** | **30.73%** | **40.95%** | **57.52%** |

**Existing methods for comparison.** We compare with existing state-of-the-art DP-trained models: DP-NFNet (De et al., 2022) and TAN (Sander et al., 2022)), both of which are trained differentially privately on ImageNet-1K using supervised learning. In addition, we present the results of several widely used *non-private* models that are pre-trained on ImageNet-1K including AlexNet (Krizhevsky et al., 2017) (supervised learning-based) and SimCLR (Chen et al., 2020a) (SSL-based) for reference. To measure the effectiveness of DP pre-training compared to synthetic pre-training, we also evaluate the model pre-trained on synthetically generated Shader21k data, denoted **(Syn)-ViP**. We also compare ViP with the non-private MAE model pre-trained on the same datasets and summarize the results in Table 7 (Appendix B.4).

## 4.2 TRANSFER LEARNING EVALUATION

To show that ViP learns high-quality representations from its training data, we evaluate its transfer learning performance on a suite of image classification tasks using both linear probing and few-shot fine-tuning. For linear probing, we use all the training samples in the downstream task training set to learn the linear classifier, while freezing all layers except for the final linear layer. For few-shot fine-tuning, we randomly select $K$ training samples from each class and fine-tune the entire model. It is worth noting that both linear probing and fine-tuning evaluations are done using *non-private* training; our pre-trained ViP model only satisfies $(\epsilon, \delta)$-DP on the LAION233M dataset.

**Linear probing.** Table 1 shows the linear probing results on four large-scale image classification datasets: ImageNet-1K, Places-365/205 and iNat-2021. The most suitable baselines in this setting are DP-NFNet and TAN, both of which are DP-trained on ImageNet-1K with $\epsilon = 8$ and represent previous state-of-the-art in large-scale DP pre-training. First of all, we find that MAE pre-training only on synthetic images (*i.e.*, (Syn)-ViP) is already comparable or even outperforms SOTA DP pre-trained models. After differentially privately pre-training on LAION233M, ViP effectively improves the performance of (Syn)-ViP on all datasets by a large margin.

Importantly, ViP even outperforms *non-private* SimCLR pre-trained on ImageNet-1K on all datasets (except ImageNet-1k itself because SimCLR does not need to transfer), and achieves similar perfor-

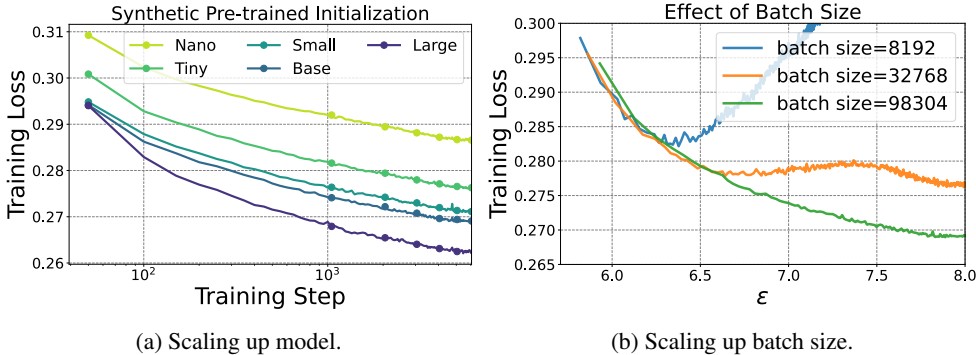

(a) Scaling up model.

(b) Scaling up batch size.

Figure 4: (**Left**) Effect of scaling up model size on MAE training loss. Larger models attain lower training loss despite the larger magnitude of noise added during DP-SGD. (**Right**) Effect of batch size on MAE training loss while fixing $\epsilon$. A large batch size is necessary for convergence.

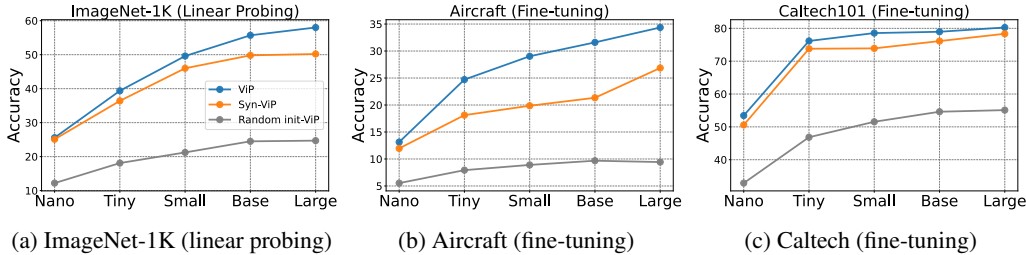

(a) ImageNet-1K (linear probing)

(b) Aircraft (fine-tuning)

(c) Caltech (fine-tuning)

Figure 5: Effect of scaling up model size on downstream performance. ViP with synthetic pre-training (blue line) benefits substantially from larger model size. In comparison, ViP with random initialization (gray line) does not benefit as much from model scaling, as the difference in performance between MAE-Large and MAE-Nano is considerably smaller.

mance as end-to-end non-privately trained AlexNet. To the best of our knowledge, this is the first time a DP-trained model can achieve similar performance on vision benchmark datasets as that of a mainstream (albeit older) model, which demonstrates the potential of our training recipe.

**Few-shot fine-tuning.** Table 2 shows the few-shot fine-tuning results on Aircraft, Caltech-101 and CIFAR-100. Similar to the linear probing result, (Syn)-ViP already outperforms TAN—the previous SOTA DP-trained model—across all evaluation settings except for 10-shot classification on Aircraft. Next, we find that ViP can largely improve upon (Syn)-ViP when the number of samples per class is small, attaining SOTA performance in all evaluation settings. ViP also achieves better performance than non-privately pre-trained AlexNet by a large margin, but falls short against non-private SimCLR despite having access to more than $100\times$ training data. Thus, our result can be viewed as both a positive and a negative result, showing that there is still a long way to go for private learning before matching the performance of mainstream vision models across the board.

## 4.3 SCALING PROPERTIES

We now study scaling properties of our training recipe, including scaling up (1) the model size, (2) the training set size, and (3) the previously known successful recipe of scaling up batch size.

**Scaling up model size.** DP-SGD training is generally unfavorable to large models because the noise magnitude increases with model size. Interestingly, we show that model performance in fact improves by scaling up model size using our training recipe. Specifically, we change the MAE-encoder size while fixing the MAE-decoder size, resulting in five different model sizes from MAE-Nano to MAE-Large; Table 4 in Appendix A.1) gives architecture details including number of parameters. All models are trained to satisfy the same $(\epsilon, \delta)$-DP guarantee with $\epsilon = 8$.

Figure 4a plots the training curve for the different-sized models. At the beginning of DP training, due to synthetic pre-training, a larger MAE model can learn more expressive features and hence the MAE training loss on LAION233M decreases as model size increases. Intriguingly, the training losses of MAE-Small/Base/Large are similar at the beginning, but larger ViT models achieve faster convergence *despite the large amount of DP noise*. Although similar observations on larger models

Table 3: Ablation studies on the effect of dataset size and batch size. The first row shows the result of (Syn)-ViP, which is the common starting point for all models in the subsequent rows. Difference in performance compared to (Syn)-ViP is shown in parentheses. See text for details. (‡ represents linear probing evaluation and ⋄ represents 10-shot fine-tuning evaluation.)

| Model | Batch Size | # Train data | Noise $\sigma$ | ImageNet-1K ‡ | Places-365 ‡ | iNat-2021‡ | Aircraft⋄ | CIFAR-100⋄ |
|---|---|---|---|---|---|---|---|---|
| *(Syn)-ViP* | - | - | - | 49.8% | 43.2% | 32.4% | 21.8% | 39.0% |
| *ViP* | 98,304 | 2M | 2.50 | 52.6% (+2.8%) | 44.8% (+1.6%) | 37.0% (+4.6%) | 29.1% (+7.3%) | 39.9% (+0.9%) |
| *ViP* | 98,304 | 23M | 0.66 | 53.7% (+3.9%) | 45.2% (+2.0%) | 37.6% (+5.2%) | 31.5% (+9.7%) | 40.5% (+1.5%) |
| *ViP* | 98,304 | 233M | 0.48 | 55.7% (+5.9%) | 46.1% (+2.9%) | 38.1% (+5.7%) | 31.6% (+9.8%) | 41.0% (+2.0%) |
| *ViP* | 8,192 | 233M | 0.41 | 43.9% (- 5.9%) | 41.0% (- 2.2%) | 27.6% (- 4.8%) | 15.0% (- 6.8%) | 39.2% (+0.2%) |
| *ViP* | 32,768 | 233M | 0.45 | 53.0% (+3.2%) | 45.1% (+1.9%) | 36.2% (+3.8%) | 30.0% (+8.2%) | 40.3% (+1.3%) |
| *ViP* | 98,304 | 233M | 0.48 | 55.7% (+5.9%) | 46.1% (+2.9%) | 38.1% (+5.7%) | 31.6% (+9.8%) | 41.0% (+2.0%) |

converge faster have also been described in the context of non-private learning (Li et al., 2020), the fact that we observe the same phenomenon in Figure 4a suggests that model scaling can be effective even for *private* learning under our training recipe.

Figure 5 shows the effect of model scaling on downstream linear probing and fine-tuning performance. In particular, the effective reduction in training loss shown in Figure 4a indeed translates to better downstream performance, with larger ViP model consistently achieving better accuracy without modifications to the training process. Moreover, comparing ViP with synthetic pre-training (blue line) vs. random initialization (gray line) shows that synthetic pre-training is crucial for unlocking this scaling behavior: the difference in performance between MAE-Large and MAE-Nano is much smaller when the model is randomly initialized.

**Scaling up dataset size.** Next, we investigate the effect of scaling up the number of training samples in ViP training. We vary the training dataset size from 2M to 23M to 233M while choosing the magnitude of injected noise $\sigma$ so that models trained on different dataset sizes satisfy $(\epsilon, \delta_n)$-DP guarantee with $\epsilon = 8$ and $\delta_n = 1/2n$, where $n$ is the number of training samples. Table 3 shows downstream evaluation results. The first row corresponds to the synthetically pre-trained ViP model and rows 2-4 correspond to DP-trained ViP models with different dataset sizes. As expected, a larger pre-training dataset size results in a higher-utility ViP model. For example, scaling from 2M to 233M gives 3.1% linear probing accuracy gain on ImageNet-1K (from 52.6% to 55.7%). Given that the collection of large labeled datasets is very costly in practice, these results highlight the significance of self-supervised learning in DP training.

**Scaling up batch size.** Scaling up the training batch size is a known effective way to achieve strong performance in DP supervised learning (Li et al., 2022a). We analyze the effect of batch size in training ViP models and show that the same observation holds for DP self-supervised learning. We consider three different batch size $B \in \{8192, 32768, 98304\}$, and keep the computational budget—number of per-sample gradient computation—the same for all batch sizes. We then select the noise $\sigma$ such that models trained with different batch size satisfy the same $(\epsilon, \delta)$-DP. As shown in Figure 4b, we find that larger batch size leads to better stability in the training process as well as faster convergence under the same computational budget. Rows 5-7 in Table 3 demonstrate that larger batch size also translates to a substantial improvement in ViP's transfer learning performance.

## 5 DISCUSSION AND FUTURE WORK

We developed a recipe for DP self-supervised learning of foundation vision models, and showed that the resulting model—ViP—can achieve downstream performance matching or exceeding that of mainstream non-private models such as SimCLR (with ImageNet-1K pre-training). Our work shows the potential of scaling DP training to internet-scale unlabeled datasets and presents several opportunities for future work. **1.** Our recipe adapted MAE to DP-SGD training with minimal modifications. It may be possible to design more specialized SSL training algorithms that conform to the requirements of DP-SGD and are more effective at learning useful representations. **2.** Multi-modal SSL is generally more effective than single-modality pre-training due to the additional supervision from cross-modal alignment (Mu et al., 2022). However, existing multi-modal SSL methods are mostly based on contrastive learning (*e.g.*, CLIP (Radford et al., 2021), SLIP (Mu et al., 2022) and FLIP (Li et al., 2022b)) and do not admit per-sample gradient computation. Recent work (Huang et al., 2023) investigated how to fine-tune CLIP on vision-language tasks with DP guarantee. Additional work may be needed to adapt these methods to DP-SGD training.

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
