# A  IMPLEMENTATION AND EVALUATION DETAILS

In this section, we provide implementation details for training and evaluating *(Syn)-ViP*, *ViP*, as well as other existing methods.

## A.1  DETAILS FOR MAE MODEL

In Table 4, we provide details for backbones of MAE model with different model sizes. Both MAE-Large and MAE-Base encoders are constructed following the identical setup described in He et al. (2022).

Table 4: Details of MAE backbone variants used in *ViP*.

| *ViP* model | MAE Backbone | Encoder depth | Encoder width | Decoder depth | Decoder width | # parameters |
|---|---|---|---|---|---|---|
| *ViP-Nano* | MAE-Nano | 12 | 192 | 4 | 512 | 18.6M |
| *ViP-Tiny* | MAE-Tiny | 12 | 384 | 4 | 512 | 34.8M |
| *ViP-Small* | MAE-Small | 12 | 576 | 4 | 512 | 61.6M |
| *ViP-Base* | MAE-Base | 12 | 768 | 4 | 512 | 99.0M |
| *ViP-Large* | MAE-Large | 24 | 1024 | 4 | 512 | 233.3M |

## A.2  DETAILS FOR VIP PRE-TRAINING

For *(Syn)-ViP* pre-training, we follow the training setup outlined in (He et al., 2022): we apply the training parameters specified in Table 8 of He et al. (2022) and pre-train pre-train *(Syn)-ViP* on the S21k dataset developed in Baradad et al. (2022), which comprises of 1,300,000 training samples, for a total of 1,000 epochs. Our *(Syn)-ViP* pre-training applies the self-supervised MAE training methodology and does not use the label information available in the S21k dataset.

We now present details for differentially private *ViP* pre-training. As mentioned in Section 3, we first initialize the model weights with *(Syn)-ViP* pre-trained on S21k dataset. Then we apply DP-AdamW[6]. See the table below for training hyperparameters.

| Model | lr ($\eta$) | warmup iterations | wd ($\lambda$) | $(\beta_1, \beta_2)$ | epsilon ($\epsilon$) | lr decay |
|---|---|---|---|---|---|---|
| *ViP-Base* | $3.84 \cdot 10^{-4}$ | 1,000 | 0.005 | (0.9, 0.95) | $10^{-8}$ | cosine |

For masking in the MAE training, we follow the random masking strategy and masking ratio of 75% in He et al. (2022) for both *(Syn)-ViP* pre-training and *ViP* pre-training. The process of executing each iteration of DP-AdamW for training the *ViP-Base* model takes approximately 25 seconds when utilizing 48 A100 (40GB) GPUs. Each epoch of the *(Syn)-ViP-Base* model's training process takes roughly 90 seconds to complete with 48 A100 (40GB) GPUs.

## A.3  DETAILS FOR DOWNSTREAM CLASSIFICATION TASK

**Linear probing.** We follow the training setup in He et al. (2022): we apply BatchNorm (Ioffe & Szegedy, 2015) before the last linear layer, and use the LARS (You et al., 2017) optimizer. We choose the base learning rate `blr` $\in \{0.1, 0.05, 0.01\}$, batch size $B = 16, 384$, weight decay $\lambda = 0.0$. We set warmup epoch as 10, and total training epoch as 90. We use the `RandomResizedCrop` and `RandomHorizontalFlip` augmentations.

**Few-shot fine-tuning.** For vision transformer based architectures, we apply the AdamW optimizer with learning rate of lr $\in \{3 \cdot 10^{-3}, 3 \cdot 10^{-4}, 3 \cdot 10^{-5}\}$ and set weight decay as 0.05. For convolutional neural networks (AlexNet, ResNet used in SimCLR), we apply the SGD optimizer because it consistently outperforms AdamW. We select learning rate lr $\in \{1 \cdot 10^{-2}, 1 \cdot 10^{-3}, 1 \cdot 10^{-4}\}$, while setting the momentum as 0.9 and the weight decay as 0.0. For all models we apply the cosine

---

[6]A variant of the standard DP-SGD — we first compute the noisy clipped stochastic gradient described in equation 3, then apply one step update of AdamW (Loshchilov & Hutter, 2017) using the estimated gradient.

learning rate decay, and use 10 warm-up epochs and fine-tine with 200 total epochs. We apply AutoAugment (Cubuk et al., 2018) for data augmentation.

## A.4 DETAILS FOR DOWNSTREAM SEGMENTATION AND DETECTION TASKS

**COCO object detection and segmentation.** We fine-tune the pre-trained *(Syn)-ViP* and *ViP* on COCO with the `Detectron2` package (Wu et al., 2019). We apply the pre-trained *(Syn)-ViP-Base* and *ViP-Base* as the ViT initializations for the detection and segmentation tasks, and apply the default hyperparameter config in `Detectron2` for ViTDet-Base.

**ADE20K semantic segmentation.** We follow the setup described in He et al. (2022) on evaluating pre-trained MAE models for semantic segmentation. We apply the UPerNet (Xiao et al., 2018) and perform fine-tuning for 100 epochs with a batch size of 16.

## A.5 DETAILS FOR DIFFERENTIALLY PRIVATE FINE-TUNING ON IMAGENET

We use the pre-trained encoders of *(Syn)-ViP* and *ViP* and apply DP-AdamW for DP end-to-end fine-tuning. The details for parameters in DP-AdamW can found in the following table.

| Model | sampling ratio $q$ | noise $\sigma$ | iterations $T$ | lr | wd |
|---|---|---|---|---|---|
| *ViP-Base / (Syn)-ViP-Base* | $262,144/n$ | 5.6 | 1,500 | $1.02 \cdot 10^{-3}$ | 0.005 |

We use 50 iterations for learning rate warm-up, and then keep the learning rate constant afterwards. For selecting parameters not presented in the aforementioned table, we adopt the default configuration of AdamW in `PyTorch` (Paszke et al., 2017). The fine-tuned model satisfies $(8, 8 \cdot 10^{-7})$-DP on the ImageNet-1K dataset in addition to the LAION233M dataset.

## A.6 DETAILS FOR FIGURE 1

For the linear probing results, we present the performance of the ViP-Large model, with the summarized results shown in the last row of Table 4. Regarding the detection and segmentation results, we utilize the ViP-Base model as the ViT backbone, and the corresponding outcomes can be found in Table 5.

## B    ADDITIONAL EXPERIMENTAL RESULTS

In this section, we provide additional experimental results on evaluating *(Syn)-ViP*, *ViP*, as well as other existing methods.

### B.1    SEGMENTATION AND DETECTION EVALUATIONS OF (SYN)-VIP/VIP

We summarize the results for object detection and segmentation in Table 5. Training details can be found in Appendix A.4.

Table 5: Evaluation of our DP models (*(Syn)-ViP*, *ViP*) as well as existing non-private baselines on COCO object detection/segmentation and ADE20K semantic segmentation.

| Model | DP? | COCO | | ADE20K |
| --- | --- | --- | --- | --- |
| | | $AP^{box}$ | $AP^{mask}$ | mIoU |
| SimCLR (Chen et al., 2020a) | ✗ | 37.9 | 33.3 | - |
| Mask R-CNN (He et al., 2017) | ✗ | 40.0 | 37.1 | - |
| RefineNet (Lin et al., 2017) | ✗ | - | - | 40.7 |
| MAE (He et al., 2022) | ✗ | 50.3 | 44.9 | 48.1 |
| *(Syn)-ViP* | ✓ | 45.0 | 40.1 | 38.8 |
| *ViP* | ✓ | 45.2 | 40.4 | 40.1 |

### B.2    ADDITIONAL EXPERIMENTS ON VIP PRE-TRAINING

In Figure 6, we plot the training loss v.s. number of training steps for *ViP* training *without (Syn)-ViP initialization*. Compared to the results in Figure 4a, when pre-training from scracth with DP-AdamW, larger models do not converge faster than smaller ones. These results further demonstrate the effectiveness of synthetic pre-training for unlocking DP-SGD training of larger vision models.

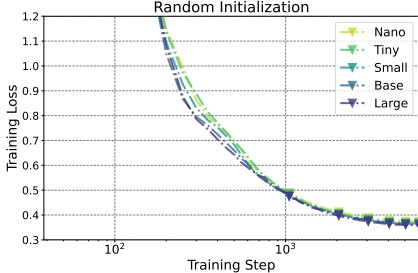

Figure 6: Training loss of different model sizes. (with random initialization).

### B.3    DP FINE-TUNING VIP ON IMAGENET-1K

Thus far, our main emphasis has been on evaluating DP pre-trained ViP through *non-private* linear probing or fine-tuning on downstream tasks. For certain use cases, the downstream task training set may be privacy-sensitive as well and DP fine-tuning is required. We simulate such a scenario by fine-tuning the privately pre-trained ViP model[7] on ImageNet-1K with DP-SGD. As a result, the fine-tuned model satisfies $(8, 8 \cdot 10^{-7})$-DP on the ImageNet-1K dataset in addition to the LAION233M dataset. We compare against prior works on training DP ImageNet models without pre-training (Kurakin et al., 2022; De et al., 2022; Sander et al., 2022); results are summarized in Table 6.

By utilizing our pre-trained *ViP* as an initialization, we observe an improvement in top-1 accuracy of more than 10% compared to the previous SOTA (Sander et al., 2022), demonstrating the efficacy of our DP pre-training recipe.

---

[7]ViP-Base pre-trained on LAION233 shown in the last row of Table 1.

Table 6: DP fine-tuning evaluation on ImageNet-1K. We compare (Syn)-ViP and ViP with existing DP training methods (DP-ResNet-18, DP-NFNet, and TAN) on ImageNet-1K.

| Model | $(\epsilon, \delta)$-DP | Top-1 Accuracy |
|---|---|---|
| DP-ResNet-18 (Kurakin et al., 2022) | $(13.2, 10^{-6})$ | 6.2% |
| DP-NFNet (De et al., 2022) | $(8, 8 \cdot 10^{-7})$ | 32.4% |
| TAN (Sander et al., 2022) | $(8, 8 \cdot 10^{-7})$ | 39.2% |
| *(Syn)-ViP* | $(8, 8 \cdot 10^{-7})$ | 48.9% $\pm$ 0.2 |
| *ViP* | $(8, 8 \cdot 10^{-7})$ | **50.3%** $\pm$ 0.3 |

## B.4 ADDITIONAL EXPERIMENTS ON THE CLASSIFICATION TASK

**Comparison with non-private MAE.** To gain a better understanding of the gap between non-private training and private training, we use the same synthetic pre-trained model as initialization and perform DP-AdamW training on LAION233M with $\sigma = 0.0$[8]. We keep most of the training parameters the same except for setting the sampling ratio to $q = 4096/n$ and the number of iterations $T = 60,000$[9]. We then evaluate the linear probing (few-shot fine-tuning) performance of the trained model and provide the results in Table 7 (Table 8).

For linear probing, our *ViP* model closes more than half the gap between the *(Syn)-ViP* model and the non-private MAE model. With a more refined training recipe, it is plausible that the gap can be reduced even further, allowing DP-trained foundation vision models to rival non-privately trained ones on certain downstream tasks. In the context of few-shot fine-tuning, a comparison between private learning and the non-private MAE model reveals considerable potential for improvement in the private learning approach.

**Comparison with ViP trained on de-duplicated LAION-2B.** Recent work has demonstrated that there exist duplicated samples in the LAION dataset, which poses copyright and privacy challenges for foundation models trained on LAION. Therefore, we also pre-train our proposed ViP model on a de-duplicated subset of LAION-2B (Schuhmann et al., 2022), denoted by Dedup-LAION-245M, which consists of a similar number of training samples (245 million) as the one we mainly consider in this work. We summarize the linear probing performance of the ViP pre-trained on Dedup-LAION-245M in Table 7. We find the ViP model pre-trained on the de-duplicated LAION achieves similar performance as the one trained on LAION-400M (Schuhmann et al., 2021).

Table 7: Linear probing evaluation on downstream classification. We compare *ViP* and *(Syn)-ViP* with (non-private) MAE (He et al., 2022).

| Model | Pre-train dataset | DP? | SSL? | ImageNet-1K[‡] | Places-365 | Places-205 | iNat-2021 |
|---|---|---|---|---|---|---|---|
| (non-private) MAE | LAION-233M | ✗ | ✓ | 60.5% | 48.3% | 51.8% | 38.5% |
| *(Syn)-ViP* | LAION-233M | ✓ | ✓ | 49.8% | 43.2% | 45.8% | 32.4% |
| *ViP* | LAION-233M | ✓ | ✓ | 55.7% | 46.1% | 48.5% | 38.1% |
| *ViP* | Dedup-LAION-245M | ✓ | ✓ | 55.5% | 46.3% | 48.1% | 38.0% |

Table 8: Fine-tuning evaluation on few-shot downstream classification. We compare *ViP* and *(Syn)-ViP* with (non-private) MAE (He et al., 2022).

| Model | Aircraft | | | Caltech-101 | | | CIFAR-100 | | |
|---|---|---|---|---|---|---|---|---|---|
| | 10-shot | 20-shot | 30-shot | 5-shot | 10-shot | 30-shot | 5-shot | 10-shot | 30-shot |
| (non-private) MAE | 36.78% | 56.82% | 66.20% | 72.93% | 84.50% | 92.78% | 34.38% | 47.98% | 62.88% |
| *(Syn)-ViP* | 21.79% | 46.85% | 58.45% | 60.51% | 76.21% | 88.48% | 27.62% | 38.96% | 55.84% |
| *ViP* | **31.62%** | **53.05%** | **64.26%** | **68.05%** | **79.03%** | **88.90%** | **30.73%** | **40.95%** | **57.52%** |

[8]In this case, the $\epsilon = +\infty$ for the $(\epsilon, \delta)$-DP.

[9]While the trained model may not necessarily achieve optimal performance, our main purpose is to present a non-private model that follows a similar training setup, with the exception of setting the noise to zero. This allows us to compare its performance to the private model.

**Linear probing evaluation of ViP with different model sizes.** We study the scaling behavior of ViP and (Syn)-ViP through linear probing. As shown in Table 10, we compare the performance of ViP and (Syn)-ViP with different model sizes. The performance of ViP consistently improves across all datasets as the model size increases. In contrast, increasing the model size from MAE-Base to MAE-Large results in less than 1% improvement in top-1 accuracy for (Syn)-ViP. These findings further underscore the effectiveness of our proposed ViP training recipe for scaling up model size in private pre-training.

Table 9: Linear probing evaluation of *ViP-LAION* with different privacy budget on ImageNet-1k classification. We vary the privacy budget epsilon ($\epsilon$) from 2.0 to $+\infty$, where our default privacy budget is $\epsilon = 8.0$ and we use $\epsilon = +\infty$ to denote the non-private MAE model.

| Model | Downstream dataset | $\epsilon = 2.0$ | $\epsilon = 4.0$ | $\epsilon = 8.0$ | $\epsilon = +\infty$ |
|---|---|---|---|---|---|
| *ViP-LAION* | ImageNet-1k | 51.4% | 53.8% | 55.7% | 60.5% |

Table 10: Additional linear probing evaluation on downstream classification (ViP with different model sizes).

| Model | # parameters | Backbone | ImageNet-1K | Places-365 | Places-205 | iNat-2021 |
|---|---|---|---|---|---|---|
| *(Syn)-ViP-S* | 61.6M | MAE-Small | 46.0% | 40.9% | 43.2% | 28.3% |
| *(Syn)-ViP-B* | 99.0M | MAE-Base | 49.8% | 43.2% | 45.8% | 32.4% |
| *(Syn)-ViP-L* | 233.3M | MAE-Large | 50.2% | 43.3% | 46.5% | 32.7% |
| *ViP-S* | 61.6M | MAE-Small | 49.6% | 42.4% | 44.7% | 30.0% |
| *ViP-B* | 99.0M | MAE-Base | 55.7% | 46.1% | 48.5% | 38.1% |
| *ViP-L* | 233.3M | MAE-Large | **58.0%** | **48.5%** | **50.8%** | **40.6%** |

## B.5 ViP ABLATION EXPERIMENTS

We study the effect of MAE-decoder depth and MAE-masking ratio in *ViP* pre-training, and evaluate different models with linear probing on ImageNet-1K. We consider the *ViP-Base* setting and the results are summarized in Table 11.

Table 11: Ablation studies on the effect of decoder depth and masking ratio in MAE.

| Model | decoder depth | masking ratio | ImageNet-1K |
|---|---|---|---|
| *ViP* (default) | 4 | 0.75 | 55.7% |
| *ViP* | *1* | 0.75 | 43.4% |
| *ViP* | *2* | 0.75 | 51.7% |
| *ViP* | *8* | 0.75 | 50.1% |
| *ViP* | 4 | *0.25* | 53.5% |
| *ViP* | 4 | *0.5* | 54.7% |