# OpenReview forum: "ViP: A Differentially Private Foundation Model for Computer Vision"
_ICLR.cc/2024/Conference — Submitted to ICLR 2024_

### Official Review · Reviewer_UcdW · 2023-10-26

**Soundness:** 2 fair
**Presentation:** 2 fair
**Contribution:** 1 poor
**Rating:** 3
**Confidence:** 3

**Summary:**

This paper focuses on differentially private representation learning for images. The authors empirically found that pretraining an MAE on synthetic images as the initialization then fine-tuning on private dataset with DP-SGD can boost the utility of learned features. Experiments show that it is even better than some non-private counterparts.

**Strengths:**

1. The differentially private representation learning is a well-motivated problem and has a wide range of applications.
2. Experimental results look superior compared to baselines.

**Weaknesses:**

1. The proposed training recipe is not new. It contains two main steps: (1) pretraining on a synthetic dataset where there is no privacy concern, then (2) fine-tuning on a private dataset with DP-SGD. Something similar was proposed in many prior papers. To name a few, [1,2] in NLP (which are also cited by the authors), and [3] in CV. A minor difference is that these works choose to pretrain on a public real dataset instead of a synthetic dataset. Therefore, I do not see much novelty in this training recipe.
2. The authors also claim that this recipe used by prior works, e.g. [1,2], is on supervised training. However, it is unclear what challenges you will have if you apply this recipe to SSL.
3. The motivation for choosing MAE is not adequately clear. There are certainly other methods that can compute gradient in a disentangled manner. Naively, the ordinary autoencoder (without mask) should also be able to do this job. Why is MAE particularly picked? If there are more options, a comparison is desired.
4. AlexNet is too old to compare, which was proposed more than 10 years ago. There are too many recent baselines you can compare. (Even SimCLR is not the latest, but at least it is within 3 years).
5. Comparison in Table 1 is not fair. It looks to me that the ViP-LAION should be ViP-ImageNet-1k so that the readers can appreciate the benefit of an additional pretraining on the synthetic dataset.
6. There are many claims in this paper without enough explanations. See my questions below.


[1] Yu, Da, et al. "Differentially Private Fine-tuning of Language Models." ICLR 2022.

[2] Li, Xuechen, et al. "Large language models can be strong differentially private learners." ICLR 2022.

[3] Luo, Zelun, et al. "Scalable differential privacy with sparse network finetuning." CVPR 2021.

**Questions:**

1. Point 1 in page 2, "...attaining high-utility learned representations requires significantly more training data...", why it is more than supervised learning?
2. Point 3 in page 2, "SSL training requires a much larger number of training epochs compared to supervised learning,..." why?
3. Still in page 2, "We also show that it is tolerant to the large amount of Gaussian noise added in DP-SGD." Where do you show and why?
4. How are your synthetic data generated? From a generative model? If so, does the training set of the generative model contain any private information?
5. At the beginning of sec 3, "1. Scaling up the number of training samples via SSL with masked autoencoder;" what does this mean?
6. At the end of sec 3.1, "With more training samples, the magnitude of the injected noise becomes smaller." Why?

---

> ### Author Response · Authors · 2023-11-18
> **Rebuttal by Authors (Part 1)**
>
> Thank you for your review. Below we attempt to resolve the questions you posed.
>
> >**Q1**: *The proposed training recipe is not new. It contains two main steps: (1) pretraining on a synthetic dataset where there is no privacy concern, then (2) fine-tuning on a private dataset with DP-SGD. Something similar was proposed in many prior papers. To name a few, [1,2] in NLP (which are also cited by the authors), and [3] in CV. A minor difference is that these works choose to pretrain on a public real dataset instead of a synthetic dataset. Therefore, I do not see much novelty in this training recipe.*
>
> **A1**: The main novelty of this training recipe is that, contrary to prior work such as [1,2], we consider training the image representation model with **self-supervised learning** rather than supervised learning. This a non-trivial effort since most SSL training algorithms, such as SimCLR, DINO, BYOL, etc. cannot be applied in a straightforward manner with DP-SGD. Through trial and error, we have identified masked autoencoder (MAE) as a suitable SSL algorithm for this purpose. As a consequence, we can now unlock training on massive internet-scaled unlabeled image datasets such as LAION, partially resolving the challenge that DP representation learning requires much more data than non-private learning [TB2020].
>
> In addition, we would like to clarify that steps (1) and (2) of our approach also differ drastically from previous work. For instance, [1,2,3] consider the case of pre-training on a large public dataset and DP fine-tuning on a small dataset for a specific task. In our setting, the synthetic pre-training step does not access any public real data, including derivatives of public data such as samples generated by a model trained on public data. Instead, we only utilize programmatically generated synthetic textures [BCW+2022], which only serves to initialize the foundation model. Subsequently, the DP pre-training step in our method aims to learn generic features about real world objects rather than specializing to a specific task, and constitutes the majority of training time for the entire pipeline.
>
> >**Q2**: *The authors also claim that this recipe used by prior works, e.g. [1,2], is on supervised training. However, it is unclear what challenges you will have if you apply this recipe to SSL.*
>
> **A2**: To reiterate, we view our training recipe as sufficiently different from the recipe used in [1,2] as mentioned in **A1**. In general, applying this recipe to SSL does require addressing several challenges, such as SSL typically requiring more training iterations which conflicts with a strong DP guarantee, the number of model parameters must be sufficiently large compared to DP fine-tuning in order to learn useful representation with DP pre-training, etc. We have detailed these challenges and how our recipe addresses them in Section 3 of our paper.
>
> >**Q3**: *The motivation for choosing MAE is not adequately clear. There are certainly other methods that can compute gradient in a disentangled manner. Naively, the ordinary autoencoder (without mask) should also be able to do this job. Why is MAE particularly picked? If there are more options, a comparison is desired.*
>
> **A3**: Indeed, we agree that there exist other SSL methods that can compute per-sample gradients. Our main contribution is the discovery, implementation and evaluation of a recipe that successfully enables large-scale differentially private SSL training for the first time, rather than presenting a comprehensive recipe that works for all SSL algorithms. In particular, we have found reconstruction-based SSL to be the most compatible with DP-SGD, and MAE has achieved superior performance compared to other methods within the reconstruction-based SSL family [HCX+2022]. With that said, other types of SSL methods can enable additional downstream capabilities, and we strongly agree that it deserves attention in future work.
>
> [TB2020] Florian Tramer and Dan Boneh. Differentially private learning needs better features (or much more data). arXiv preprint arXiv:2011.11660, 2020.
>
> [HCX+2022] Kaiming He, Xinlei Chen, Saining Xie, Yanghao Li, Piotr Dollár, and Ross Girshick. Masked autoencoders are scalable vision learners. In Proceedings of the IEEE/CVF Conference on Computer Vision and Pattern Recognition, pp. 16000–16009, 2022.
>
> [BCW+2022] Manel Baradad, Richard Chen, Jonas Wulff, Tongzhou Wang, Rogerio Feris, Antonio Torralba, and Phillip Isola. Procedural image programs for representation learning. Advances in Neural Information Processing Systems, 35:6450–6462, 2022.

---

> ### Author Response · Authors · 2023-11-18
> **Rebuttal by Authors (Part 2)**
>
> >**Q4**: *AlexNet is too old to compare, which was proposed more than 10 years ago. There are too many recent baselines you can compare. (Even SimCLR is not the latest, but at least it is within 3 years).*
>
> **A4**: We do not intend to compare against AlexNet or SimCLR as baselines, but rather as non-private reference points to measure our progress against. In particular, our ViP model significantly outperforms all existing differentially private (DP) baseline models by a large margin. The main goal of our work is to show that DP representation learning can be done by scaling up the training data and leveraging SSL. This is in stark contrast to the result in previous work [TB2020, DBH+2022] that showed representations of differentially private deep learning models are even worse than handcrafted representations. The following excerpt from [TB2020] supports this claim concretely: **“Our results show that private deep learning remains outperformed by handcrafted priors on many tasks, and thus has yet to reach its “AlexNet moment””**. Thus, it is more appropriate to view the non-DP result as the maximum attainable model performance rather than as a baseline.
>
> >**Q5**: *Comparison in Table 1 is not fair. It looks to me that the ViP-LAION should be ViP-ImageNet-1k so that the readers can appreciate the benefit of an additional pretraining on the synthetic dataset.*
>
> **A5**: Per your suggestion, we have conducted new experiments on ImageNet – we train our ViP-base model on ImageNet that is (eps=8)-differentially private. As shown in the new comparison table (Table 1), ViP trained on ImageNet-1k underperforms ViP trained on LAION400m.  This is to be expected, as ViP-LAION utilizes much more data than both DP supervised learning baselines and ViP-ImageNet-1k. In fact, we view this as the fundamental difference between our work and prior studies, as our DP SSL recipe enables the use of internet-scale unlabeled datasets for the purpose of learning generic image representations.
>
> >**Q6**: *Point 1 in page 2, "...attaining high-utility learned representations requires significantly more training data...", why it is more than supervised learning?*
>
> **A6**: This is a claim made in [TB2020], please refer to [TB2020] and A4 for more information.
>
> >**Q7**: *Point 3 in page 2, "SSL training requires a much larger number of training epochs compared to supervised learning,..." why?*
>
> **A7**: This is a common practice in SSL, where SSL methods usually require more training iterations than standard supervised training. For example, on ImageNet-1K, supervised learning usually needs ~100 epochs, whereas SSL methods need >500 epochs. Please refer to [CKN+2020, HCX+2022] for more details.
>
> >**Q8**: *Still in page 2, "We also show that it is tolerant to the large amount of Gaussian noise added in DP-SGD." Where do you show and why?*
>
> **A8**: To clarify, in Figure 3(a), we showed that by scaling the training set size, it is possible to optimize the training loss even when a large amount of Gaussian noise has been added in DP-SGD. This is due to a drastic reduction in the effective noise, as explained below in **A11**.
>
> >**Q9**: *How are your synthetic data generated? From a generative model? If so, does the training set of the generative model contain any private information?*
>
> **A9**: The synthetic data only contains programmatically generated textures using the method described in [BCW+2022] and does not correspond to any real world images. Please refer to Fig. 2 for examples of generated synthetic image samples.
>
> >**Q10**: *At the beginning of sec 3, "1. Scaling up the number of training samples via SSL with masked autoencoder;" what does this mean?*
>
> **A10**: Sorry this phrasing may have been unclear. We are referring to our strategy of using the self-supervised learning technique of masked autoencoder to unlock the use of internet-scaled datasets such as LAION.
>
> >**Q11**: *At the end of sec 3.1, "With more training samples, the magnitude of the injected noise becomes smaller." Why?*
>
> **A11**: This is due to the inherent privacy-utility trade-off in DP-SGD. For example, as described in Theorem 1 in [ACG+2016], when the number of training samples ($N$) is large, we can utilize a larger batch size $B$ while fixing the number of training steps, thus the effective noise magnitude ($\sigma/B$) becomes smaller.
>
>
> [CKN+2020] ​​T. Chen, S. Kornblith, M. Norouzi, and G. Hinton. A simple framework for contrastive learning of visual representations. In International conference on machine learning, pages 1597–1607. PMLR, 2020a.
>
> [ACG+2016] Martin Abadi, Andy Chu, Ian Goodfellow, H Brendan McMahan, Ilya Mironov, Kunal Talwar, and Li Zhang. Deep learning with differential privacy. In Proceedings of the 2016 ACM SIGSAC conference on computer and communications security, pp. 308–318, 2016.

---

> ### Comment · Reviewer_UcdW · 2023-11-18
>
> I sincerely thank the authors for their detailed rebuttal. I'll give my comments based on the response.
> 1. To A1 and A2:
>     + In terms of novelty, I understand that reconstruction-based SSL is different from supervised learning, but I disagree that steps (1) and (2) are drastically different from prior works. Step (1) in both prior work and this work aims to warm up the training with non-sensitive data (synthetic in this work vs. public in theirs), while step (2) in both prior and this work aims to continue training on a private dataset with DP-SGD. The difference the authors mention stems from the representation learning itself, rather than the training recipe. This recipe in spirit, in my point of view, has a large overlap with prior works (e.g. [1,2,3] in my review above).
>     + I agree that DP SSL is non-trivial, but I disagree that this work made non-trivial contribution in addressing this problem. Challenges for enabling DP training with most existing SSL methods, e.g. SimCLR, BYOL, as the authors also mentioned, arise due to the fact that they cannot compute gradients in a disentangled manner. However, this work does not genuinely solve this challenge, instead, the authors propose to bypass this problem by turning to reconstruction-based SSL, where the per-example gradient is available. So let me be more clear, I wonder if there is any non-trivial challenge for running DP-SGD on reconstruction-based SSL methods.
>     + Speaking of SSL, this work also overclaims on the contribution towards DP SSL, because the authors only solve reconstruction-based DP SSL. Readers like me who expect to see constrastive methods might be disappointed.
> 2. To A3: I admire the authors' honesty in admitting that MAE is chosen by trial and error in A1. However, I am not convinced MAE is the optimal solution down the road.
> 3. To A4: thanks for the clarification.
> 4. To A5: thanks for the update. The last row in Table 1 now is more appropriate to compare with prior works.
> 5. To A6-10: thanks for the clarification. I suggest either adding pointers to references or explanations (as in rebuttal) in the draft to minimize confusion for readers.

---

> > ### Author Response · Authors · 2023-11-20
> >
> > Although the idea of applying DP-SGD to MAE training sounds straightforward to implement in hindsight, there are several engineering challenges that arise from the use of massive datasets such as LAION.
> > - Implementation of distributed gradient clipping and aggregation. Even though there are packages such as torch.DistributedDataParallel that implement distributed non-private training, adopting them for DP training is highly non-trivial. The gradient clipping operation is very computationally expensive, which, coupled with the fact that we have to pre-train the model on the 233M LAION dataset for 3 epochs, demands that this operation is as efficient as possible. Due to the extremely large batch size, we also had to implement gradient accumulation and aggregation and test them rigorously to make sure everything is correct. Note that all of this has to be done for each SSL method we consider, and any minor modification has to be tested carefully as well, so the cost of exploration is extremely high.
> > - Efficient implementation of Poisson subsampling. This is actually a major issue with scaling DP-SGD to large datasets, because random disk access (from a distributed set of machines no less) is inherently slow. This issue is further exacerbated by the way LAION stores data into zip files of approx. 10000 samples each, so each random access is also associated with an unzip operation. We had to reformat the LAION dataset and write a custom data loader to make training feasible, reducing the per-iteration time **from 28.8s to 0.8s**. Please keep in mind that many papers that perform DP training **do not do this properly** [KSC+2022, DBH+2022], and instead permute the whole dataset every epoch and then draw batches of samples in fixed order. This means that such works do not directly benefit from theoretically rigorous and strong amplification by subsampling (we are not aware of any theoretical alternative analysis that could apply in such cases).
> >
> > We did not discuss these challenges in the paper because they do not contribute to its main message, but we can definitely add some of these challenges to the main body in the final version.
> >
> > More importantly, we strongly believe that the merit of a paper should be evaluated based on how much it contributes to the existing literature. To this end, our paper benefits the DP research community as it proposes, implements and evaluates a new way for DP representation learning via SSL, showing that it can lead to significantly better quality of learned representation. And thus, we believe our paper is pushing forward the limits of private learning and is worthy of acceptance. The fact that our training recipe seems simple and straightforward is, in our view, a desirable outcome, because it means that any reader familiar with DP can easily understand why data scaling helps, how SSL is the way forward for DP training, and what kind of result can one hope to achieve when adopting this approach. We will also release our model and code so that our arduous work can benefit the entire research community, towards making DP foundation models a reality rather than a farfetched concept. We hope that our work leads to a whole ecosystem of privately trained foundation models as a counterpart to the non-private ones that exist right now.
> >
> > ​​[KSC+2022] Alexey Kurakin, Shuang Song, Steve Chien, Roxana Geambasu, Andreas Terzis, Abhradeep Thakurta. Toward Training at ImageNet Scale with Differential Privacy. arXiv:2201.12328.
> >
> > [DBH+2022] Soham De, Leonard Berrada, Jamie Hayes, Samuel L. Smith, Borja Balle. Unlocking High-Accuracy Differentially Private Image Classification through Scale. arXiv:2204.13650.

---

### Official Review · Reviewer_MY6t · 2023-10-31

**Soundness:** 2 fair
**Presentation:** 2 fair
**Contribution:** 1 poor
**Rating:** 3
**Confidence:** 3

**Summary:**

The paper proposes how to train vision foundation models with privacy using the framework of differential privacy. The work targets only a single version of self-supervised encoders, namely MAE. The motivation is that the encoders can still leak private information. The ViP pre-trained encoder achieves accuracy for linear probing of 55.7% on ImageNet, which is comparable with AlexNet.

**Strengths:**

1. The paper proposes a DP method for the large-scale encoder models.

**Weaknesses:**

1. The motivation is very poor: the encoders should not be trained on copyright or private data in the first place instead of preventing the detection that they were trained on such data.
2. The method is limited to only the MAE and state-of-the-art methods such as DINO or DINO v2 are not supported, let alone the contrastive-based encoders such as SimCLR.
3. The performance of the encoder trained on the large data is only 55.7% for the linear probing on ImageNet. One does not have to go through the same huge effort but instead, use a publicly trained model such as AlexNet to obtain the same performance.

Other comments:
1. Figure 1 is misplaced. First of all, it is way too early, since even no reference to the Figure is given on page 1. Second, the comparison is a strawman argument. ViP should be compared with the corresponding MAE encoder trained without DP.
2. It is claimed that: "More recently, Meehan et al. (2023) showed that non-generative vision SSL models can also be probed to reveal sensitive information about individual samples in its training data when given partial information." However, this work does not address the issues in the SSL encoders from Meehan et al. (2023), where no MAE encoders were considered!
3. " However, most vision SSL training algorithms are based on contrastive learning, where the objective function
depends on multiple samples in an entangled manner". This is not correct. There are many non-contrastive SSL methods, for example, SimSiam [1], DINOv1 [2], or DINO v2 [3], which is a state-of-the-art SSL encoder. In general, [4] considers contrastive and non-contrastive encoders.

**References:**
1. "Exploring Simple Siamese Representation Learning". Xinlei Chen, Kaiming He. https://arxiv.org/abs/2011.10566 CVPR 2021.
2. "Emerging Properties in Self-Supervised Vision Transformers" https://openaccess.thecvf.com/content/ICCV2021/papers/Caron_Emerging_Properties_in_Self-Supervised_Vision_Transformers_ICCV_2021_paper.pdf
3. "DINOv2: Learning Robust Visual Features without Supervision" https://arxiv.org/abs/2304.07193
4. "Contrastive and Non-Contrastive Self-Supervised Learning Recover Global and Local Spectral Embedding Methods" https://arxiv.org/pdf/2205.11508.pdf

**Questions:**

See the Weaknesses above.

---

> ### Author Response · Authors · 2023-11-18
> **Rebuttal by Authors (Part 1)**
>
> Thank you for your review. Below we attempt to resolve the questions you posed.
>
> >**Q1**: *The motivation is very poor: the encoders should not be trained on copyright or private data in the first place instead of preventing the detection that they were trained on such data.*
>
> **A1**: We respectfully disagree with this statement. The current legal landscape (e.g. GDPR) considers anonymized and aggregated data as non-personal data, and thus does not prohibit processing of such data. There remain questions as to how this notion of anonymization and aggregation can be applied to ML, but DP currently stands out as the gold standard for privacy protection and has been adopted by organizations such as the US Census Bureau, Google, Apple, etc. [Des21, CDE+2023]. The goal of our paper is to push the boundary of DP foundation model training and bring this gold standard notion closer to current practice in ML.
>
> >**Q2**: *The method is limited to only the MAE and state-of-the-art methods such as DINO or DINO v2 are not supported, let alone the contrastive-based encoders such as SimCLR.*
>
> **A2**: Indeed, we focused on reconstruction-based learning (MAE) due to its compatibility with DP-SGD, as discussed in Section 3.1. Our work is meant to adapt reconstruction-based learning to enable large-scale differentially private SSL training for the first time, rather than presenting a comprehensive recipe that works for all SSL algorithms. With that said, contrastive learning or other types of SSL methods (e.g., DINO) can enable additional downstream capabilities, and we strongly agree that it deserves attention in future work. We agree that our work only handles one category of self-supervised learning in computer vision -- we will clarify and emphasize further in the final version.
>
> >**Q3**: *The performance of the encoder trained on the large data is only 55.7% for the linear probing on ImageNet. One does not have to go through the same huge effort but instead, use a publicly trained model such as AlexNet to obtain the same performance.*
>
> **A3**: We agree that one can easily achieve 55.7% linear probing accuracy on ImageNet using other publicly available image encoders. However, the purpose of our work is not to compete with non-DP models, but rather to show that DP representation learning can be done by scaling up the training data and leveraging SSL. This is in stark contrast to the result in previous work [TB2020, DBH+2022] that showed representations of differentially private deep learning models are even worse than handcrafted representations. Thus, it is more appropriate to view the non-DP result as the maximum attainable model performance rather than as a baseline. In practice, however, a portion of the training data may be publicly accessible. In such situations, a natural way to apply our findings is to initialize the model by non-DP training on the public data, then fine-tune with DP on the private data.
>
> >**Q4**: *Figure 1 is misplaced. First of all, it is way too early, since even no reference to the Figure is given on page 1. Second, the comparison is a strawman argument. ViP should be compared with the corresponding MAE encoder trained without DP.*
>
> **A4**: Thank you for your suggestions on presentation. As Figure 1 is mentioned in the first section, we would like to highlight the empirical results on the first page. Nevertheless, we are happy to move Figure 1 to the second page in our camera-ready version. In Figure 1, our goal is to demonstrate that ViP is comparable with several existing popular non-private models. We will add the MAE encoder results in Table 1 in our camera-ready version.
>
> [Des21] Damien Desfontaines. A list of real-world uses of differential privacy. https://desfontain.es/privacy/real-world-differential-privacy.html, 10 2021. Ted is writing things (personal blog).
>
> [CDE+2023] Rachel Cummings, Damien Desfontaines, David Evans, et al. Challenges towards the Next Frontier in Privacy. arXiv preprint arXiv:2304.06929.
>
> [TB2020] Florian Tramer and Dan Boneh. Differentially private learning needs better features (or much more data). arXiv preprint arXiv:2011.11660, 2020.
>
> [DBH+2022] Soham De, Leonard Berrada, Jamie Hayes, Samuel L Smith, and Borja Balle. Unlocking high- accuracy differentially private image classification through scale. arXiv preprint arXiv:2204.13650, 2022.

---

> ### Author Response · Authors · 2023-11-18
> **Rebuttal by Authors (Part 2)**
>
> >**Q5**: *It is claimed that: "More recently, Meehan et al. (2023) showed that non-generative vision SSL models can also be probed to reveal sensitive information about individual samples in its training data when given partial information." However, this work does not address the issues in the SSL encoders from Meehan et al. (2023), where no MAE encoders were considered!*
>
> **A5**: There seems to be a misunderstanding. Our intention of citing Meehan et al. (2023) is to argue that embedding models trained with SSL, even if they do not contain decoders, can reveal sensitive information about its training data. In the case of MAE, the decoder is readily available and we suspect it is even easier to extract sensitive information from the model. Of course, further investigation is required to verify this claim, but we believe it gives sufficient motivation for studying how to train SSL embedding models, such as MAE, with differential privacy. We will clarify this point in the revision.
>
> >**Q6**: *" However, most vision SSL training algorithms are based on contrastive learning, where the objective function depends on multiple samples in an entangled manner". This is not correct. There are many non-contrastive SSL methods, for example, SimSiam [1], DINOv1 [2], or DINO v2 [3], which is a state-of-the-art SSL encoder. In general, [4] considers contrastive and non-contrastive encoders.*
>
> **A6**: Thank you for pointing out the references, we will add and discuss these references in our updated version. In our revised version, we will discuss these approaches as a second category when discussing SSL methods and differentially private training. Please note that the four approaches mentioned above cannot be directly applied with differentially private training. In particular, SimSiam [1] requires BatchNorm, DINOv1/DINOv2 requires centering operation (e.g., centering operation in Algorithm 1 of DINOv1 [2]), and the loss function used [4] depends on multiple samples (thus cannot compute per-sample gradient).  With that said, we strongly agree that ‘how to train non-reconstruction based SSL models with DP’ deserves attention in future work.

---

### Official Review · Reviewer_kYFB · 2023-11-07

**Soundness:** 4 excellent
**Presentation:** 4 excellent
**Contribution:** 3 good
**Rating:** 8
**Confidence:** 4

**Summary:**

The authors proposed VIP, a recipe of privately training vision foundation models through self-supervised learning. The main insights are two-fold: 1) masked autoencoder is a suitable algorithm for DP-SGD which allows per-sample gradient clipping, as opposed to contrastive learning; 2) warm-start through non-private synthetic pretraining can greatly accelerate the training. The authors conducted comprehensive experiments to demonstrate the effectiveness of VIP, showing that it surpasses state-of-the-art methods on a variety of learning tasks.

**Strengths:**

- The motivation and insights are adequately delivered
- Strong and comprehensive empirical results
- Good writing and presentation

**Weaknesses:**

I don't see any apparent weakness of this work, but only a few minor suggestions:
- The experiments should be running over multiple random seeds, and please include the standard deviations in the tables as well
- It would be better to include a non-private version of VIP (i.e., non-private MAE) for comparison in the experiments (to reflect the cost of DP)
- It would be better to emphasize in the title and abstract that this paper focuses on applying DP to SSL, which is different from most prior works that focus on applying DP to supervised learning
- There is a concurrent work [1] which applied DP to the continued pretraining CLIP (using batched gradient clipping). The authors should discuss this work in Section 5
- Section 2, Eq. equation 1 -> Eq. (1)


Reference

[1] Huang, Alyssa, et al. "Safeguarding Data in Multimodal AI: A Differentially Private Approach to CLIP Training." arXiv preprint arXiv:2306.08173 (2023).

**Questions:**

I don't have further questions.

---

> ### Author Response · Authors · 2023-11-18
> **Rebuttal by Authors**
>
> Thank you for your review. Below we attempt to resolve the questions you posed.
>
> >**Q1**: *The experiments should be running over multiple random seeds, and please include the standard deviations in the tables as well.*
>
> **A1**: Thank you for your suggestions, we will update our main tables with standard deviations in our camera-ready version.
>
> >**Q2**: *It would be better to include a non-private version of VIP (i.e., non-private MAE) for comparison in the experiments (to reflect the cost of DP).*
>
> **A2**: We have included the results of non-private ViP (i.e., MAE) models in Table 7 and Table 8 (Appendix B.4). Per your suggestion, we have highlighted the results in our updated version.
>
> >**Q3**: *It would be better to emphasize in the title and abstract that this paper focuses on applying DP to SSL, which is different from most prior works that focus on applying DP to supervised learning.*
>
> **A3**: Thank you for your suggestion on presentation. Per your suggestion, we have highlighted the self-supervised learning in our updated abstract, i.e., 'In this work, we propose as a mitigation measure a recipe to train foundation vision models via self-supervised learning with differential privacy (DP) guarantee.'
>
> >**Q4**: *There is a concurrent work [1] which applied DP to the continued pretraining CLIP (using batched gradient clipping). The authors should discuss this work in Section 5.*
>
> **A4**: Thank you for pointing out related work [HLN+2023]. We would like to highlight that, similar to previous work [YNB+2021, LTL+2022], [HLN+2023] applied public pre-trained models and studied how to fine-tune the pre-trained models with DP. In contrast, our work investigated how to pre-train vision foundation models with differential privacy, without accessing any public real-world data. We have added [HLN+2023] to Section 5 and discussed pre-training CLIP with DP for future work in our revision.
>
> >**Q5**: *Section 2, Eq. equation 1 -> Eq. (1).*
>
> **A5**: Thank you for pointing this out, we have fixed this in our updated version.
>
>
> [HLN+2023] Alyssa Huang, Peihan Liu, Ryumei Nakada, Linjun Zhang, Wanrong Zhang. "Safeguarding Data in Multimodal AI: A Differentially Private Approach to CLIP Training." arXiv preprint arXiv:2306.08173 (2023).
>
> [LTL+2022] Xuechen Li, Florian Tramer, Percy Liang, and Tatsunori Hashimoto. Large language models can be strong differentially private learners. In International Conference on Learning Representations, 2022a.
>
> [YNB+2021] Da Yu, Saurabh Naik, Arturs Backurs, Sivakanth Gopi, Huseyin A Inan, Gautam Kamath, Janardhan Kulkarni, Yin Tat Lee, Andre Manoel, Lukas Wutschitz, et al. Differentially private fine-tuning of language models. arXiv preprint arXiv:2110.06500, 2021.

---

> > ### Comment · Reviewer_kYFB · 2023-11-18
> >
> > Thanks for the response. I have read through other reviews and responses, and decided to keep my score. Overall I believe this paper makes solid contributions towards training DP foundation models. I will vote for acceptance.

---

### Meta-Review · Area_Chair_a2gk · 2023-12-12

**Metareview:**

(a) Summarize the scientific claims and findings of the paper based on your own reading and characterizations from the reviewers.

This paper is motivated by three shortcomings of current practice in differentially private SSL for large vision models. First, DP training requires a large number of samples. This is due ot the fact that the sensitivity, in general, goes down with the batchsize (and the sample size). Successful training with DP hinges on having large number of samples. Secondly, a more popular approach to SSL, which is contrastive learning, does not scale well when it comes to private training. This is because the gradients are tangled together across the minibatch and it is not immediately clear how to handle clipping. Oftentimes lossy methods like microbatching is used to handle it at the expense of adding more noise then what might be necessary. Lastly, SSL from scratch requires a large number of iterations, and the combined loss in privacy can be intolerable. Aa a remedy, the paper proposes MAE (Masked AutoEncoder), where the training loss can be separated out for each sample, and warming up the training with synthetics data.

(b) What are the strengths of the paper?

Several important observations are made. In Figure 3, the study on the number of pretraining samples and synthetic pretraining epochs show consistent gains. The main experimental results show the gain of using large pretrianing data compared to other DP training methods when compared in linear probing tasks.

(c) What are the weaknesses of the paper? What might be missing in the submission?

The motivation is not convincing enough. Why do we need the pretraining to satisfy differential privacy? This is critical as it determines what kind of privacy we need to enforce, most notably what should be the unit of privacy? This is a very complex and subtle issue not yet resolved by the community, especially so when copyright is involved. The usual sample level DP enforced in this paper has little meaning when it comes to copyright. If we want to be strict, one is supposed to protect the whole "book", for example. Not each sentence in the book. This is very complex and the paper is not expected to address this issue, but wanted to point it out.

Perhaps more important related issue is that I believe that LAION400M is not deduplicated. Meaning there are examples that are copies or variations of one another. For example, Abbas et al. ("Semdedup: Data-efficient learning at web-scale through semantic deduplication,") propose SemDeDup, which starts with the CAT-filtered LAION-440M subset, further employing clustering to remove semantic duplicates. This is not just a simple matter of running a deduplication as a preprocessing, as at the heart of the proposed approach is internet scale data collection. Imagine a blog post that is shared and copied multiple times. Perhaps also quoted and modified before being shared. Since such data collection is one crucial part of the proposed approach, care is needed in both deduplicating and deciding the unit of privacy. Note how this is crucial in the proposed scenario where internet scale data is necessary, as opposed to the more popular scenario of private finetuning where the private data is curated and controlled.

Given the popularity of the contrastive losses in SSL, one could imagine that the proposed method is trading off the choice of MAE for better handle of DPSGD. This tradeoff is not clearly characterized, as experiments and comparisons involve multiple parts (MAE, different dataset, differing data size).

But perhaps more fundamentally, most of novelty in this paper comes from the fact that the paper is addressing a scenario less popular (the private pretraining model), and not the solution proposed. One could argue that all the components of this paper has been around in existing literature.

**Justification For Why Not Higher Score:**

To summarize the weaknesses,

- the paper combines solutions that exist in literature
- the exact unit of privacy and deduplication have not been addressed enough
- the tradeoffs between the choice of loss and implementation of clipping in DPSGD is not carefully characterized.

**Justification For Why Not Lower Score:**

N/A

---

### Decision · Program_Chairs · 2024-01-16

Reject